# Prognostic Value of Chromatin Structure Typing in Early-Stage Non-Small Cell Lung Cancer

**DOI:** 10.3390/cancers15123171

**Published:** 2023-06-13

**Authors:** Luning Mao, Jianghua Wu, Zhongjie Zhang, Lijun Mao, Yuejin Dong, Zufeng He, Haiyue Wang, Kaiwen Chi, Yumeng Jiang, Dongmei Lin

**Affiliations:** 1Key Laboratory of Carcinogenesis and Translational Research (Ministry of Education), Department of Pathology, Peking University Cancer Hospital & Institute, Beijing 100142, China; maoluning@bjmu.edu.cn (L.M.); fjctwjh@126.com (J.W.); hyuewang@163.com (H.W.); chikaiwen@bjmu.edu.cn (K.C.); 18818271915@139.com (Y.J.); 2Department of Epidemiology, School of Public Health, University of Pittsburgh, Pittsburgh, PA 15261, USA; zhz152@pitt.edu; 3My-BioMed Technology (Guangzhou) Co., Ltd., Guangzhou 510000, China; lijun.mao@my-biomed.com (L.M.); yuejin.dong@my-biomed.com (Y.D.); jennifer.he@my-biomed.com (Z.H.)

**Keywords:** NSCLC, DNA ploidy, nucleotyping, adjuvant therapy, disease-free survival

## Abstract

**Simple Summary:**

In this work, we evaluated the prognostic value of the chromatin structure in patients with early-stage lung cancer. We assessed the associations of DNA ploidy, nucleotyping, and tumor–stroma ratio (TSR) with 5-year disease-free survival rates. Clarifying whether patients with homogeneous and heterogeneous chromatin can benefit from adjuvant chemotherapy can guide the decision-making regarding chemotherapy after lung cancer resection, improve the survival rate of patients, and reduce the incidence and cost of adverse events related to lung treatment.

**Abstract:**

(1) Background: Chromatin structure typing has been used for prognostic risk stratification among cancer survivors. This study aimed to ascertain the prognostic values of ploidy, nucleotyping, and tumor–stroma ratio (TSR) in predicting disease progression for patients with early-stage non-small cell lung cancer (NSCLC), and to explore whether patients with different nucleotyping profiles can benefit from adjuvant chemotherapy. (2) Methods: DNA ploidy, nucleotyping, and TSR were measured by chromatin structure typing analysis (Matrix Analyser, Room4, Kent, UK). Cox proportional hazard regression models were used to assess the relationships of DNA ploidy, nucleotyping, and TSR with a 5-year disease-free survival (DFS). (3) Results: among 154 early-stage NSCLC patients, 102 were non-diploid, 40 had chromatin heterogeneity, and 126 had a low stroma fraction, respectively. Univariable analysis suggested that non-diploidy was associated with a significantly lower 5-year DFS rate. After combining DNA ploidy and nucleotyping for risk stratification and adjusting for potential confounders, the DNA ploidy and nucleotyping (PN) high-risk group and PN medium-risk group had a 4- (95% CI: 1.497–8.754) and 3-fold (95% CI: 1.196–6.380) increase in the risk of disease progression or mortality within 5 years of follow-up, respectively, compared to the PN low-risk group. In PN high-risk patients, adjuvant therapy was associated with a significantly improved 5-year DFS (HR = 0.214, 95% CI: 0.048–0.957, *p* = 0.027). (4) Conclusions: the non-diploid DNA status and the combination of ploidy and nucleotyping can be useful prognostic indicators to predict long-term outcomes in early-stage NSCLC patients. Additionally, NSCLC patients with non-diploidy and chromatin homogenous status may benefit from adjuvant therapy.

## 1. Introduction

Lung cancer is a leading cause of cancer-related death worldwide and the second most diagnosed cancer among both men and women [1]. An estimated 2.2 million people were diagnosed with lung cancer worldwide in 2020, among which non-small cell lung cancer (NSCLC) accounted for about 85% of the total cases [1]. Despite the significant advances in predictive biomarkers and cancer therapies in recent decades, the prognosis of NSCLC patients is still of concern [2]. A recent epidemiologic study using the US Cancer Statistics database suggested that the 5-year survival rates of patients with stage I and stage II NSCLC are 68.4% and 45.1%, respectively, let alone those with metastatic NSCLC [2]. Similarly, dismal prognostic outcomes are also observed in early-stage NSCLC patients in China, with an estimated 50% of stage II NSCLC patients surviving within 5 years after surgical resection [3]. Approximately 20–40% of stage IA-IIB NSCLC patients are prone to develop local or distant recurrence after complete resection [4,5].

Genomic instability has been found to be associated with chromatin reorganization, suggesting that features of chromosomal structure, including DNA ploidy, nucleotyping, and stroma fraction, can be used as predictive biomarkers for prognostic outcomes. In many tumors, abnormal chromosomes are also indicative of poor prognoses [6]. Higher-order chromatin structure regulates gene expression during cell differentiation [7], and the mutation frequency of tumor cells is also affected by chromatin organization [8,9]. Studies have demonstrated that chromatin structure typing can stratify the prognostic risk of various cancers, including colorectal cancer, gynecological cancer, and prostate cancer, exhibiting stable performance [10]. However, there is very little research focusing on the prognostic value of chromosomal structure in NSCLC.

In addition, biomarkers guide the selection of clinical treatment regimens. Surgical resection is recommended as the first choice for stage I NSCLC, and postoperative adjuvant chemotherapy, radiotherapy, and targeted drug therapy are not recommended after the complete resection of stage IA-IIB lung cancer. For IA-IIB stage NSCLC after complete resection, approximately 20–40% of patients may develop local or distant recurrence [4,5]. Postoperative adjuvant chemotherapy can increase the 5-year survival rate of stages II–IIIa NSCLC patients by 4–15% [11,12,13], but there is currently no consensus on whether postoperative adjuvant chemotherapy is necessary for stage I NSCLC patients. In the clinical trial of CALGB9633, although chemotherapy improved the overall survival of patients with stage IB NSCLC, the improvement was not statistically significant; however, for patients with tumor diameters larger than 4 cm, chemotherapy significantly improved the overall survival (OS) and disease-free survival (DFS) (HR = 0.69, *p* = 0.043) [14]. Other studies have also shown that adjuvant chemotherapy failed to improve the OS and DFS of patients with stage IB NSCLC, even patients with visceral pleural invasion and other factors [15,16,17,18]. In recent years, a number of genes related to the prognosis and chemotherapy efficacy of NSCLC have been found, such as ERCC1 (excision repair cross complementary gene 1), RRM1 (ribonucleotide reductase M 1), BRCA1 (breast cancer susceptibility gene 1), TS (thymidylate synthase), and p53 protein [19,20,21,22,23]. However, no prospective trials have been conducted to demonstrate the utility of these markers in predicting the efficacy of adjuvant chemotherapy. The European Society for Medical Oncology does not recommend the use of these routine clinical biomarkers to judge prognosis, guide therapeutic protocol, and predict the efficacy of chemotherapeutic drugs [24]. Therefore, there is an urgent need to identify new biomarkers for prognostic risk stratification and treatment strategy selection to improve the long-term outcomes for early-stage NSCLC patients.

This study aims to evaluate the prognostic value of the chromatin structure in patients with early-stage NSCLC by assessing the associations of DNA ploidy, nucleotyping, and tumor–stroma ratio (TSR) with 5-year DFS rates. Clarifying whether patients with chromatin homogeneous and heterogeneous tumors can benefit from adjuvant chemotherapy can guide decision-making of chemotherapy after NSCLC resection, improve the survival rate of patients, and reduce the incidence and cost of adverse events related to NSCLC treatment.

## 2. Materials and Methods

### 2.1. Study Population

A total of 154 stage I or stage II NSCLC patients who underwent a lobectomy with regional lymph node dissection at Peking University Cancer Hospital from January 2013 to December 2017 were included in this study. We retrospectively reviewed their clinical records and pathological specimens to confirm the diagnosis according to the 8th edition of the primary tumor, lymph node, and metastasis (TNM) classification of the American Joint Commission of Cancer. All patients included in this analysis had complete clinical data on age, gender, histological type, TNM stage, and adjuvant therapy. This study was approved by the ethics committees of the Peking University Cancer Hospital (No. 2020KT18). Written consent was obtained from all participants.

### 2.2. DNA Image Cytometry

Formalin-fixed paraffin-embedded (FFPE) specimens meeting the following study requirements were obtained: 1. the number of pathological specimens matched the clinical patient information. 2. The specimens included 2 paraffin sections with a thickness of 50 μm and 2 paraffin sections with a thickness of 5 μm. 3. The content of tumor tissues in the sections was not less than 50%, and the proportion of necrotic tissues was less than 10%. 4. Paraffin tissues were not stained. One 5 μm section was used for haematoxylin-eosin (H&E) staining to determine the area of the enclosing tumor. The tumor area in the 50 μm section was removed according to the circled tumor area. After gradient ethanol deparaffinization, cells were filtered through a 60 μm nylon mesh filter after digestion with 0.5 mg/mL proteinase VIII. After removing the supernatant, the filtered cells were resuspended in a phosphate buffer, and a 100 μL suspension was absorbed for centrifugal smear. The smear was air-dried and fixed with formaldehyde, followed by Fergen staining. The stained nuclear coating was scanned using a high-resolution digital pathology scanner (Aperio AT2, Leica, Wetzlar, Germany). The DNA ploidy, nucleotyping, and TSR of each patient were obtained by analyzing the four dimensions of entropy value (GLEM-4D) calculated by the size of the epithelial nuclei, the gray value of each pixel point, and different sampling windows using chromatin structure typing analysis (Matrix Analyser, Room4, Kent, UK).

### 2.3. Biomarker Measurements

#### 2.3.1. DNA Ploidy

The Fergen-stained nuclei were automatically divided into tumor nuclei, reference nuclei, and discarded nuclei groups. DNA ploidy histograms were created using a PWS classifier (Matrix Analyser, Room4, Kent, UK), with the integrated optical density (IOD) of the nucleus. With lymphocyte nuclei as internal diploid controls, DNA ploidy histograms were categorized into four groups: diploid, aneuploid, tetraploid, and polyploid; where, aneuploid, tetraploid, and polyploid samples were classified as non-diploid.

#### 2.3.2. Nucleotyping

Nuclear coating scan images were analyzed using a chromatin structure typing analysis instrument (Matrix Analyser, Room4, Kent, UK) to obtain the chromatin value of each patient. Patients with the chromatin value ≥ 0.044 were defined as the chromatin homogeneity (CHO) group and otherwise as the chromatin heterogeneity (CHE) group.

#### 2.3.3. TSR

The H&E-stained sections were scanned by a 40× digital tomography scanner (Aperio AT2, Leica, Wetzlar, Germany) with an image resolution of 1.82 μm/pixel. Pathologists used software tools (Matrix Analyser, Room 4, Kent, UK) to mark tumor areas on scanned images. The stromal fraction of selected tumor areas was automatically calculated by the software (Matrix Analyser, Room 4, Kent, UK). With a threshold of 0.50, samples were divided into 2 groups: low interstitial (<0.50) and high interstitial (≥0.50).

These three biomarkers were then grouped into combined indicators, among which non-diploid DNA, CHE, and high TSR were considered risk factors for poor prognoses. As a result, three levels of categorical variables were created for DNA ploidy and nucleotyping panel (PN), the DNA ploidy and TSR panel (PS), and nucleotyping and TSR panel (NS) separately. In each panel, subjects with zero, one, or two risk factors were categorized as low-, intermediate-, and high-risk subjects. In addition, we also combined all three biomarkers into one predictor variable representing the overall DNA ploidy, nucleotyping status, and TSR (PNS) profile. The low-, intermediate-, and high-risk subgroups for PNS were subjects with zero, one or two, and three risk factors, respectively.

### 2.4. Outcome Assessment

The primary outcome of this study was the 5-year DFS after surgical resection. Patients who did not experience tumor recurrence or metastasis were censored at the time of last contact or 60 months, whichever came first. The evaluation of the patient’s disease progression was up until August 2022.

### 2.5. Statistical Analysis

Descriptive analysis was performed to summarize the demographic and clinical characteristics of these 154 patients. Continuous variables are presented as the means ± standard deviations, and categorical variables are reported as frequencies and percentages. The differences in baseline variables were examined using the two-sample *t* test, the chi-square test, or Fisher’s exact test, as appropriate.

DFS rates were estimated by the Kaplan–Meier method, and Kaplan–Meier curves with log-rank estimates were used to depict time-to-event parameters. The hazard ratio (HR) and 95% confidence interval (CI) were obtained for each individual association of DNA ploidy, nucleotyping, and TSR with the 5-year DFS rate after surgery. The associations of the combined biomarkers PN, PS, NS, and PNS with the 5-year DFS rate were also assessed. Multivariable Cox proportional hazards regression models were used to examine these associations after adjusting for confounding factors. We also performed stratified analysis and interaction tests using Cox models to determine whether these associations changed with the adjuvant therapy status.

Stata 16.0 (Stata Corp., College Station, TX, USA) and IBM SPSS Statistics for Windows 22.0 (IBM Corp., Armonk, NY, USA) were used for all statistical analyses, and a two-tailed *p* < 0.05 was considered statistically significant.

## 3. Results

### 3.1. Patient Characteristics

Table 1 summarizes the demographic and clinical characteristics of the 154 patients included in this study. The mean age at the time of surgery was 61.76 years (range 35–81 years), and more than half of the included patients were male (61.0%) and nonsmokers (51.3%). Nearly 70% of early-stage NSCLC patients were in stage I, with the majority diagnosed as adenocarcinoma (59.7%). The proportions of stage I and II NSCLC patients receiving adjuvant therapy, including postoperative adjuvant chemotherapy, radiotherapy, and targeted drug therapy, were 19.4% and 60.8%, respectively.

There was a significant difference in the histological types of NSCLC between the DNA diploid and non-diploid subgroups, where more than two-thirds of patients with a non-diploid DNA status were diagnosed as adenocarcinoma (ρ = −0.226, *p* = 0.005). Similarly, patients in the CHE group had a higher tendency to suffer from adenocarcinoma (77.5%) than those in the CHO group (ρ = −0.215, *p* = 0.008). A positive correlation between DNA ploidy and nucleotyping was also observed (ρ = 0.392, *p* < 0.001). CHE was predominantly observed in the non-diploid subgroup, and only one NSCLC patient with CHE was identified in the DNA diploid subgroup (Table 1).

### 3.2. Prognostic Values of Individual Biomarkers

During a median follow-up period of 44.2 months, 47 patients experienced tumor recurrence or metastasis. The 5-year DFS rate of early-stage NSCLC patients with non-diploidy DNA was 23.90% lower than that of patients with diploidy DNA (HR = 3.006, [95% CI: 1.402–6.443], *p* = 0.005). We did not find a significant correlation between nucleotyping or TSR and 5-year DFS rates. After controlling for potential confounders, the risk of tumor progression in early-stage NSCLC patients with non-diploid DNA was 3.2 times higher than that in those with diploid DNA 5 years after surgical intervention (adjusted hazard ratio (AHR) = 3.215, [95% CI: 1.462–7.073], *p* = 0.004).

We further explored these correlations stratified by histological types of NSCLC. The effect of non-diploid DNA on the risk of tumor progression in adenocarcinoma patients was consistent with that in the overall cohort (AHR = 10.761, [95% CI: 1.427–81.183], *p* = 0.021), while the negative correlation between the CHE and 5-year DFS rate became significant after adjusting for confounders (AHR = 3.208, [95% CI: 1.464–7.029], *p* = 0.004). No statistically significant correlation was observed between all three biomarkers and the primary outcome in squamous cell carcinoma patients (Figure 1).

### 3.3. Prognostic Values of Combined Biomarkers

Table 2 presents the unadjusted and adjusted correlations between the combined biomarkers and the 5-year DFS rate. After combining DNA ploidy and nucleotyping indicators for prognostic risk stratification, the PN low-, intermediate-, and high-risk groups had 51, 64, and 39 subjects, respectively. The 5-year DFS rate in the PN low-risk group was significantly higher than that in the PN intermediate-risk group (HR = 2.682, [95% CI: 1.198–6.003]) and PN high-risk group (HR = 3.226, [95% CI: 1.377–7.557]). These significant correlations remained unchanged after controlling for potential confounding factors. There was also evidence suggesting a growing rusk of tumor progression as the increase of risk stratification of DNA ploidy and nucleotyping status (*p* = 0.004) (Appendix A).

We also examined the correlation between the PS combined with NS and the 5-year DFS after surgery among early-stage NSCLC patients. In the PS panel, the PS low-risk group was associated with a significantly improved 5-year DFS rate compared with the PS intermediate-risk group (AHR = 2.499, [95% CI: 1.150–5.429]) but not with the PS high-risk group (AHR = 2.390, [95% CI: 0.840–6.800]) after adjusting for covariates. No statistically significant correlation was found between the NS combined biomarker and the long-term survival outcome in early-stage NSCLC patients.

In addition, we further grouped all three biomarkers into one explanatory variable, in which patients were categorized into three groups based on the number of risk factors in their PNS profile. The majority of patients (N = 102, 67.5%) were assigned to the PNS intermediate-risk group, whereas the PNS low- and high-risk groups consisted of 43 and 6 patients, respectively. The PNS intermediate- and high-risk groups had a 2- (AHR = 2.444, [95% CI: 1.088–5.493]) and 4-fold (AHR = 4.312, [95% CI: 1.242–14.964]) increased risk of tumor progression or mortality at 5 years after surgery compared to the PNS low-risk group. Higher PNS risk factors were also associated with impaired 5-year survival outcomes (*p* = 0.009). The results were similar when we restricted the study population to adenocarcinoma patients, with the exception that we observed a statistically significant correlation of the combined biomarker of PS and NS with the 5-year DFS rate (Figure 2 and Appendix A). Additionally, we found a positive correlation between the PN status and tumor differentiation (ρ = 0.257, *p* = 0.013). However, tumor differentiation, as well as the presence of mutations (EGFR, KRAS, etc.) for adenocarcinoma were not significant predictors for DFS (Appendix A).

### 3.4. Prognostic Values of Individual and Combined Biomarkers Stratified by Postoperative Adjuvant Therapy

Figure 3 shows that 103 early-stage NSCLC patients did not receive adjuvant therapy and 51 patients received postoperative adjuvant therapy. Patients with or without adjuvant therapy experienced similar rates of disease progression or mortality by the end of the 5-year follow-up (33.45% vs. 33.30%, HR = 1.138, [95% CI: 0.627–2.067], *p* = 0.627).

Patients with CHE who received adjuvant therapy seemed to have improved 5-year DFS rates compared to those who did not receive adjuvant therapy (HR = 0.226, [95% CI: 0.050–1.013]), and patients with CHO who did not receive adjuvant therapy had higher risks of tumor recurrence, metastasis, or mortality (HR = 2.028, [95% CI: 1.002–4.102]); although both associations were only marginally significant. There was evidence indicating a significant interaction between the PN risk subgroups and adjuvant therapy (*p*-interaction = 0.021), which was mainly driven by the protective association between adjuvant therapy and the enhanced 5-year DFS rate in PN high-risk patients (HR = 0.214, [95% CI: 0.048–0.957]) (Figure 4). However, no statistically significant correlation was found between adjuvant therapy and long-term survival outcome in other biomarker subgroups.

## 4. Discussion

DNA ploidy, nucleotyping, and TSR have important implications in predicting disease progression risks and patient treatment outcomes in many epithelial cancers. In this study, we first evaluated the clinical characteristics and prognostic values of DNA ploidy, nucleotyping, and TSR alone or in combination in early-stage NSCLC patients. Then we discussed the performance of DNA ploidy and nucleotyping in predicting the curative effect of postoperative adjuvant therapy.

Previous studies have shown that non-diploidy promotes the development of tumors, although the mechanism by which the DNA-ploidy status of tumor tissues affect the prognosis of patients has not been well elucidated [25]. Kildal et al. [26] identified non-diploidy as a marker of poor prognosis for patients with leiomyosarcoma and adenosarcoma. Sheltzer et al. [27] found that the instability of the aneuploid genome may lead to the invasive growth of advanced malignant tumors with complex nucleotypes. Consistently, our study hypothesized that non-diploidy was also a poor prognostic factor for early-stage NSCLC patients. Incorporating histological types of NSCLC into analysis, the non-diploidy DNA status was demonstrated to exert a pernicious influence on DFS in early-stage lung adenocarcinoma. In contrast, we found no association between DFS and DNA ploidy in lung squamous cell carcinoma. The mechanistic relationship between non-diploidy DNA status and poor prognosis might be due to the instability of the genome caused by aneuploidy.

Previous studies have proven the independent prognostic significance of chromatin structure typing in the recurrence-free survival rate of early-stage ovarian cancer [28,29,30]. Among high-risk patients with stage II colon cancer, the DFS of CHO patients was significantly higher than that of the CHE patients [31]. In the present study, we found that nucleotyping can be used as an independent prognostic factor for DFS in adenocarcinoma. The 5-year DFS of patients with CHE was 14.51% and 29.81% lower than that of patients with CHO in NSCLC and adenocarcinoma, respectively. In contrast, nucleotyping did not show a significant prognostic prediction effect on patients with lung squamous cell carcinoma, and the 5-year DFS was even worse in lung squamous cell carcinoma patients with CHO. This might be due to the different histological origins between lung adenocarcinoma and lung squamous cell carcinoma. Future research will be directed at a large sample size and mechanism research to explain the correlation between nucleotyping and histological types of lung cancer.

Ploidy represents the content of DNA, while nucleotyping represents the change in chromatin structure, both of which reflect the change in DNA. Previous studies have found a correlation between the two biomarkers, with CHO patients are more likely to have diploid phenotypes, while non-diploid patients are also more likely to have CHE [31]. Our data also suggested a correlation between the two biomarkers at the cellular level. As recently reported, the combination of these two markers could predict the prognosis of patients with stage II colorectal cancer [32]. We confirmed, for the first time, the independent prognostic value of the combination of DNA ploidy and nucleotyping for DFS in early-stage NSCLC, and the 5-year DFS rates in the low-risk group, intermediate-risk group, and high-risk group were 81.75%, 61.30%, and 54.14%, respectively. Our results revealed that the combination of DNA ploidy and nucleotyping can serve as a universal marker for cancer recurrence or metastasis in early-stage NSCLC.

The TSR based on H&E-staining tissue sections is used to measure the ratio of surrounding stroma to tumor tissue. The matrix provides tumors with growth factors, cytokines, and metabolites to stimulate angiogenesis, leading to tumorigenesis and inducing epithelial–mesenchymal transition (EMT) [29]. Therefore, the high content of stromal tumor may represent the metastatic phenotype of tumor cells. Previous results have confirmed that high stroma is associated with an increased risk of recurrence and a low survival rate in colorectal cancer and prostate cancer [31,33,34,35,36,37]. In our study, the stroma was not suitable for prognosis analysis in the early-stage NSCLC, probably due to the influence of pulmonary carbon deposition on the quantification of the stroma ratio by the automatic analysis tool.

According to the Guidelines for the Diagnosis and Treatment of Primary Lung Cancer (2018 Edition) of the Chinese Society of Clinical Oncology (CSCO), adjuvant chemotherapy is generally not recommended for patients with stage IIA NSCLC who have undergone complete resection. Although adjuvant chemotherapy has been recommended for patients with stage I NSCLC, postoperative factors include poorly differentiated tumors, vascular invasion, wedge resection, tumor diameter >4 cm, visceral pleural involvement, and lymphadenopathy (NCCN, 2019V7). However, in a meta-analysis, patients with stage IA disease (n = 347) were found to have a worse prognosis after chemotherapy (HR = 1.40; 95% CI, 0.95–2.06) [38]. Therefore, a more effective prognostic indicator is needed to determine the prognosis of patients with early-stage NSCLC and to guide the follow-up treatment.

Postoperative adjuvant therapy did not improve DFS in 154 patients with early-stage NSCLC in this study. After grouping patients by stage or lymph node metastasis status, the improvement of DFS was not significant in stage II patients with lymph node metastasis (Figure 3). However, when patients were stratified by nucleotyping and PN, postoperative adjuvant therapy significantly improved DFS in patients with CHE or high-risk factors for PN. In contrast, patients with CHO or PN low and intermediate risks had a lower DFS rate after postoperative adjuvant therapy. On the basis of these results, we can infer that stratification of patients according to ploidy and nucleotyping will help select early-stage NSCLC patients who may benefit from postoperative adjuvant therapy. Since PN high-risk patients can benefit from adjuvant therapy, we suggest a sufficient treatment duration or aggressive postoperative treatment. As no DFS benefit of adjuvant therapy is observed in NSCLC patients with PN intermediate- and low-risks in the current study, routine follow-up may be sufficient.

## 5. Conclusions

In conclusion, the combination of ploidy and nucleotyping is a useful prognostic indicator to predict the recurrence and metastasis of early-stage NSCLC. In addition, patients with these two high-risk factors can benefit from adjuvant therapy. In future studies, the combination of ploidy-nucleotyping can be further evaluated as a predictive biomarker to guide chemotherapy decision-making, and its integration with routine pathological examinations will contribute to the clinical decision-making in the treatment of early NSCLC patients.

## Figures and Tables

**Figure 1 cancers-15-03171-f001:**
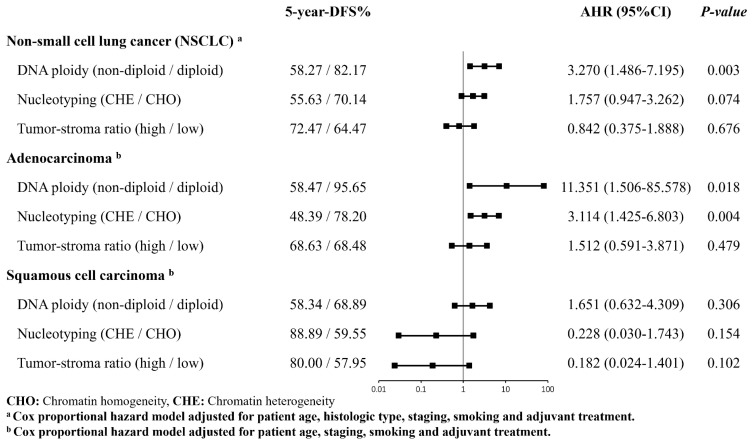
Multivariate analysis of ploidy, nucleotyping, and TSR as standalone predictors for DFS.

**Figure 2 cancers-15-03171-f002:**
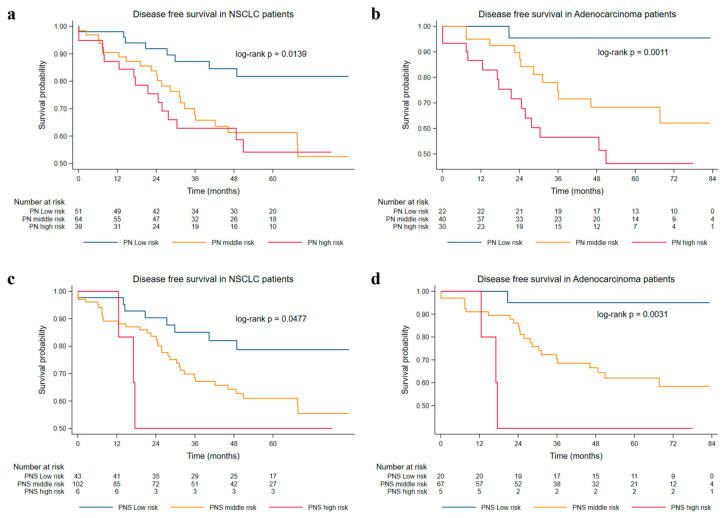
The 5-year DFS in ploidy-nucleotyping (PN) and ploidy-nucleotyping-tumor–stroma ratio (PNS). Kaplan–Meier curves of patients with NSCLC or adenocarcinoma grouped by PN and PNS. (**a**) Comparison of DFS in patients with NSCLC among the PN low-risk group, PN intermediate-risk group, and PN high-risk group. (**b**) Comparison of DFS in patients with adenocarcinoma among the PN low-risk group, PN intermediate-risk group, and PN high-risk group. (**c**) Comparison of DFS in patients with NSCLC among the PNS low-risk group, PNS intermediate-risk group, and PNS high-risk group. (**d**) Comparison of DFS in patients with adenocarcinoma among the PNS low-risk group, PNS intermediate-risk group, and PNS high-risk group.

**Figure 3 cancers-15-03171-f003:**
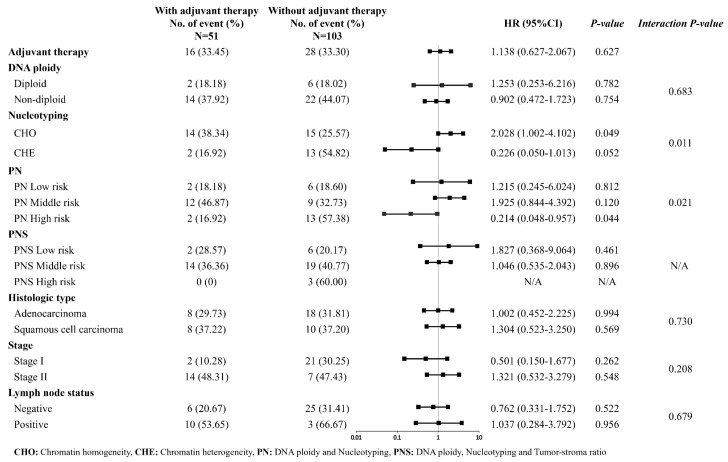
The 5-year DFS after surgery in NSCLC patients with or without postoperative adjuvant therapy, classified by ploidy and nucleotyping.

**Figure 4 cancers-15-03171-f004:**
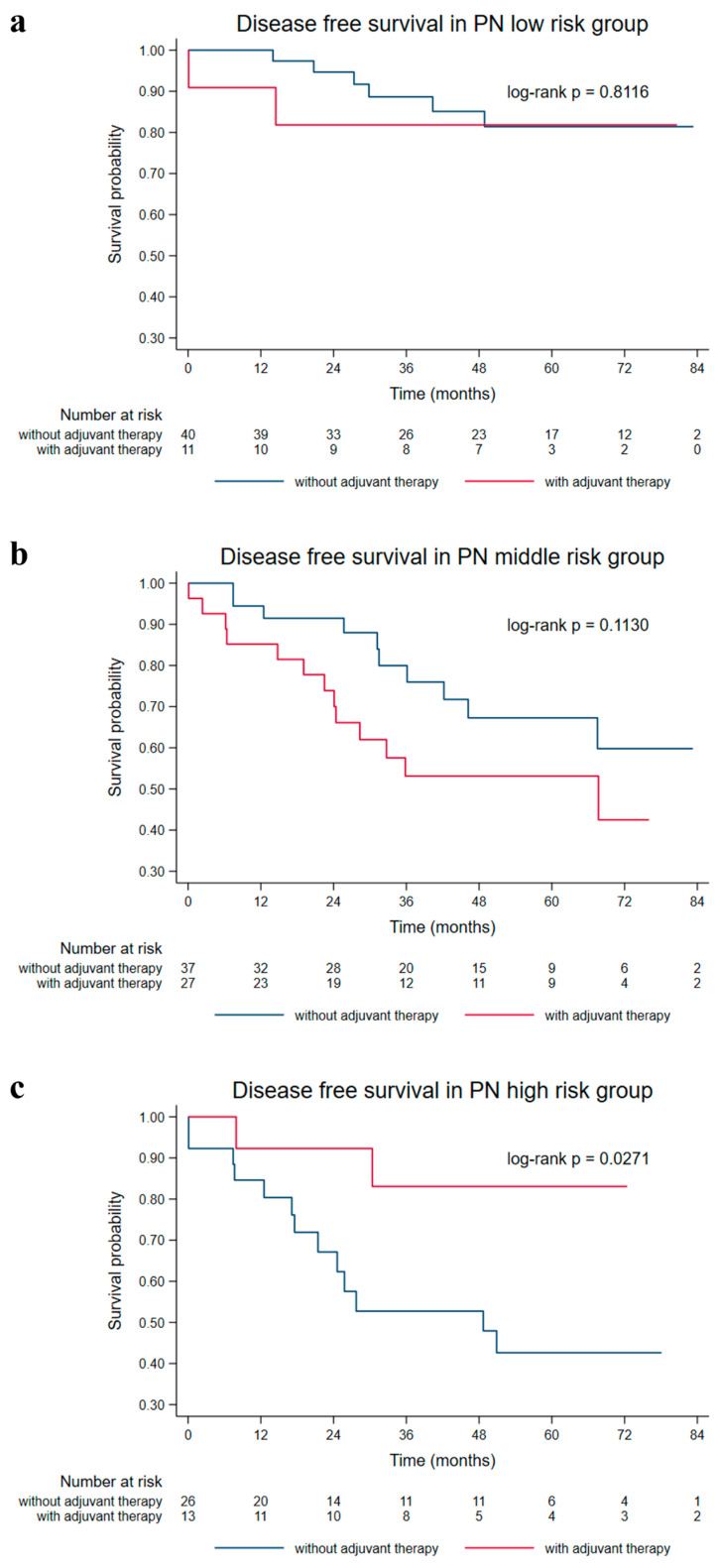
The 5-year DFS in early-stage NSCLC patients with and without adjuvant therapy based on ploidy-nucleotyping (PN) status. Kaplan–Meier plots illustrating DFS in early-stage NSCLC patients with and without adjuvant therapy based on the ploidy-nucleotyping (PN) status. (**a**) DFS in PN low-risk patients. (**b**) DFS in PN intermediate-risk patients. (**c**) DFS in PN high-risk patients.

**Table 1 cancers-15-03171-t001:** The correlations of DNA ploidy and nucleotyping with relevant clinical pathological variables.

Variables	n (%)	Diploid (%)	Non-Diploid (%)	Coefficient	*p* Value	CHO (%)	CHE (%)	Coefficient	*p* Value
Age, year		61.5 ± 8.25	61.89 ± 9.19	0.021	0.796	61.67 ± 8.30	62.03 ± 10.39	0.018	0.827
Gender				0.092	0.255			0.104	0.198
Male	94 (61.0)	35 (67.3)	59 (57.8)			73 (64.0)	21 (51.5)		
Female	60 (39.0)	17 (32.7)	43 (42.2)			41 (36.0)	19 (47.5)		
Histological type				−0.226	0.005			−0.215	0.008
Adenocarcinoma	92 (59.7)	23 (42.2)	69 (67.6)			61 (53.5)	31 (77.5)		
Squamous cell carcinoma	62 (40.3)	29 (55.8)	33 (32.4)			53 (46.5)	9 (22.5)		
Stage				0.094	0.244			−0.008	0.923
Stage I	103 (66.9)	38 (73.1)	65 (63.7)			76 (66.7)	27 (67.5)		
Stage II	51 (33.1)	14 (26.9)	37 (36.3)			38 (33.3)	13 (32.5)		
pT stage				0.014	0.697			−0.066	0.349
pT1	98 (63.6)	34 (65.4)	64 (62.7)			71 (62.3)	27 (67.5)		
pT2	44 (28.6)	13 (25.0)	31 (30.4)			32 (28.1)	12 (30.0)		
pT3	12 (7.8)	5 (9.6)	7 (6.9)			11 (9.6)	1 (2.5)		
Lymph node status				0.159	0.049			0.105	0.194
Negative	126 (81.8)	47 (90.4)	79 (77.5)			96 (84.2)	30 (75.0)		
Positive	28 (18.2)	5 (9.6)	23 (22.5)			18 (15.8)	10 (25.0)		
Smoking				−0.128	0.111			−0.192	0.017
No	79 (51.3)	22 (42.3)	57 (55.9)			52 (45.6)	27 (67.5)		
YES	75 (48.7)	30 (57.7)	45 (44.1)			62 (54.4)	13 (32.5)		
Recurrence or metastasis				0.235	0.003			0.122	0.132
No	107 (69.5)	44 (84.6)	63 (61.8)			83 (72.8)	24 (60.0)		
YES	47 (30.5)	8 (15.34)	39 (38.2)			31 (27.2)	16 (40.0)		
DNA ploidy				N/A	N/A			0.392	<0.001
Diploid	52 (33.8)	N/A	N/A			51 (44.7)	1 (2.5)		
Non-diploid	102 (66.2)	N/A	N/A			63 (55.3)	39 (97.5)		
Nucleotyping				0.392	<0.001			N/A	N/A
CHO	114 (74.0)	51 (98.1)	63 (61.8)			N/A	N/A		
CHE	40 (26.0)	1 (1.9)	39 (38.2)			N/A	N/A		
TSR				0.086	0.289			−0.025	0.757
LS	126 (81.8)	44 (84.6)	82 (80.4)			92 (80.7)	34 (85.0)		
HS	25 (16.2)	6 (11.5)	19 (18.6)			19 (16.7)	6 (15.0)		
Carbon deposition	3 (2.0)	2 (3.9)	1 (1.0)			3 (2.6)	0 (0.0)		
Total	154 (100)	52 (100)	102 (100)			114 (100)	40 (100)		

CHO: Chromatin homogeneity, CHE: Chromatin heterogeneity, TSR: Tumor–stroma ratio, LS: Low stroma, HS: High stroma.

**Table 2 cancers-15-03171-t002:** Univariate and multivariate analyses of combined factors as predictors of DFS in early-stage NSCLC patients.

Variables	n	5-Year-DFS% (95% CI)	HR (95% CI)	*p*-Value	AHR (95% CI) ^a^	*p* Value	*p* for Trend
PN							
Diploid and CHO (PN low risk)	51	81.75 (66.51–90.52)	Ref.		Ref.		
Diploid and CHE or non-diploid and CHO (PN intermediate risk)	64	61.30 (46.66–73.04)	2.682 (1.198–6.003)	0.016	2.763 (1.196–6.380)	0.017	0.004
Non-diploid and CHE (PN high risk)	39	54.14 (34.89–69.93)	3.226 (1.377–7.557)	0.007	3.601 (1.497–8.754)	0.004	
PS							
Diploid and LS (PS low risk)	44	79.28 (62.54–89.16)	Ref.		Ref.		
Diploid and HS or non-diploid and LS (PS intermediate risk)	88	58.82 (46.06–69.54)	2.499 (1.150–5.429)	0.021	2.803 (1.244–6.314)	0.013	0.053
Non-diploid and HS (PS high risk)	19	64.78 (37.34–82.58)	2.038 (0.735–5.649)	0.171	2.390 (0.840–6.800)	0.102	
NS							
CHO and LS (NS low risk)	92	67.32 (55.53–76.63)	Ref.		Ref.		
CHO and HS or CHE and LS (NS intermediate risk)	53	65.02 (48.26–77.54)	0.945 (0.505–1.769)	0.859	1.001 (0.527–1.900)	0.998	0.387
CHE and HS (NS high risk)	6	50.00 (11.09–80.37)	2.053 (0.704–5.987)	0.188	2.192 (0.747–6.431)	0.153	
PNS							
0 high-risk factor (PNS low risk)	43	78.72 (61.61–88.85)	Ref.		Ref.		
1 or 2 high-risk factors (PNS intermediate risk)	102	60.96 (49.29–70.73)	2.200 (1.020–4.746)	0.045	2.444 (1.088–5.493)	0.031	0.009
3 high-risk factors (PNS high risk)	6	50.00 (11.09–80.37)	3.782 (1.117–12.812)	0.033	4.312 (1.242–14.964)	0.021	

PN: DNA ploidy and nucleotyping, PS: DNA ploidy and tumor–stroma ratio, NS: Nucleotyping and tumor–stroma ratio, PNS: DNA ploidy, nucleotyping, and tumor–stroma ratio. CHO: Chromatin homogeneity, CHE: Chromatin heterogeneity, LS: Low stroma, HS: High stroma. ^a^ Cox proportional hazard model adjusted for patient age, histological type, stage, smoking, and adjuvant therapy.

## Data Availability

The original data supporting the conclusions of our article will be available from the corresponding authors upon reasonable request.

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
