# Peer review of "Prognostic Value of Chromatin Structure Typing in Early-Stage Non-Small Cell Lung Cancer"

_cancers, 2023, doi:10.3390/cancers15123171_

Round 1

Reviewer 1 Report

This manuscript described unique prognostic parameters regarding nuclear chromatin structure of patients with non-small cell lung carcinoma (NSCLC). The results of this study are interesting and would provide special interest to journal readers. However, in my opinion, this manuscript had some problematic points.

Major points.

1.      No references or evidence were scattered within the manuscript. For example, in the beginning of this manuscript or Introduction, the authors described as “Lung cancer is a leading cause of death and the second most common cancer in both men and women globally.” (page 2, lines 42-43), and as “Despite the discovery of predictive biomarkers and advancement of cancer therapy over the past few decades, the prognosis of NSCLC patients is still of concern.” (page 2, lines 45-47). These sentences were not accompanied with references, and seems not to be evidence-based. Similar portions lacking references or evidence were numerous in this text. Such presentation does not respect the elementary rules of a scientific writing.

2.      The authors described as “The purpose of this study is to evaluate the prognostic value of chromatin structure in patients with early lung cancer by analysis.” (page 2, lines 87-88), and as “Clarifying whether patients with chromatin … after lung cancer resection, improve the survival rate of patients … to lung cancer treatment” (page 2, lines 90-93). I believe that this study aimed to evaluate clinicopathological features of NSCLC, but “lung cancer” including small cell carcinoma.

3.      How does “tumor-stroma ratio” relate to chromatin structure typing?

4.      This manuscript should be edited by native English scientist.

Minor points:

1.      Abrupt abbreviations: “NSCLC” (Abstract); “OS” (page 2, line 74); “DFS” (page 2, line 74);

2.      Possibly second decimal place of percentage display is not required in parentheses of Table 1.

3.      Squamous cell carcinoma, not “squamous carcinoma” (Page 6, Table 1; page 7, line 213; page Figure 1; page 13, line 306; …)

4.      Not abbreviated as “CHO” or “CHE”: “chromatin heterogenous” (page 13, line 314); “chromatin homogenous” (page 13, line 315); “chromatin homogeneity” (page 13, line 325), and “chromatin heterogeneity” (page 13, lines 326-327).

5.      Repeated explanation for abbreviation: “tumor-stroma ratio (TSR)” (page 2, line 89); “tumor-stroma ratio (TSR)” (page 3, lines 120-121); “Tumor-stroma ration (TSR)” (page 3, line 138); “tumor-stroma ratio (TSR)” (page 7, line 203) and “Tumor-stroma ratio (TSR)” (page 13, line 336).

6.      Possibly unnecessary abbreviations: “ESMO” (page 2, line 82); “AJCC” (page 3, line 100); “TNM” (page 3, line 101; “LMS” (page 12, line 300); “AS” (page 12, line 300).

7.      Different abbreviations: “hematoxylin-eosin (H&E)” (page 3, line 112) and “HE-stained” (page 13, line 336).

This manuscript should be edited by native English scientist.

Author Response

Dear editors and reviewers,

Thank you for your letter and for the reviewers’ comments on our manuscript entitled “Prognostic Value of Chromatin Structure Typing in Early-stage Non-small Cell Lung Cancer” (ID: cancers-2407168). All of these comments were very helpful for revising and improving our paper. We have studied these comments carefully and have made corresponding corrections that we hope will meet with your approval. [The changes in the revised manuscript are marked in red.]* The responses to the reviewers’ comments are provided below.

We would like to express our great appreciation to you and the reviewers for the comments on our paper. If you have any further queries, please do not hesitate to contact us.

Kind regards,
Luning Mao

Reviewer 2 Report

1. Figure 3 is of very poor quality and needs to be redone.

2. The authors did not evaluate the influence of the degree of tumor differentiation, as well as the presence of mutations (EGFR, KRAS, etc.) for adenocarcinoma and their relationship with the prognosis for different Ploidy-Nucleotyping (PN) status?

3. To what extent was the lung resection performed? Has a lymph node dissection been performed? Can this information be added to the analysis?

Author Response

(The authors gave the same response as above.)

Round 2

Reviewer 1 Report

The amended version of the manuscript has not been fully revised according to my previous comments and still has some problematic points.

1.       Although the authors replied as “response minor points: These have now been corrected as suggested.” within “Response to Reviewer 1 Comments”, inappropriate portions still remain in this amended version. For example:

1)      Abrupt abbreviations (previously minor points 1):

tumorTSR (revised version, page 2, line 90).

2)      Repeated explanation for abbreviation (previous minor points 5):

    Tumou-stroma ratio (TSR) [revised version, page 3, line 139]; tumor-stroma ratio (TSR) [revised version, page 7, line 203-204]

2.       Misprint: Tumou (revised version, page 3, line 139)

3.       No spaces between words and bracket parentheses: “…globally[1].” (page 2, line 43); “… cancern[2]” (page 2, line 46); ….

I believe that these above points could be adequately corrected if the manuscript had been properly edited by native English scientists.

I have a question as to whether the revised manuscript has really been edited in such a way. At least, it would be disrespectful and dishonest for scientists to submit a revised manuscript in which the problematic pints I have suggested is still present.

This manucript should be properly edited by true native English scientists.

Author Response

Thanks for your careful review and professional suggestions. Acting on your suggestions, we have rechecked the language issues in the entire manuscript and consulted a native language polishing personnel for language polishing. Please refer to the revised manuscript for further review.

Reviewer 2 Report

I have no more comments on the article. I think that in its present form the manuscript can be accepted for publication.

Author Response

Thank you for your recognition of our article, and we will continue to work hard. Thank you again.

Round 3

Reviewer 1 Report

The manuscript has been amended adequately, and I cannot find any additional problems.

I believe that this revised version can be accepted for publication in Cancers.